# Possible Etiological Factors for the Development of Molar Incisor Hypomineralization (MIH) in Austrian Children

**DOI:** 10.3390/dj12030044

**Published:** 2024-02-20

**Authors:** Sarra Altner, Ivan Milutinovic, Katrin Bekes

**Affiliations:** 1Department of Paediatric Dentistry, School of Dentistry, Medical University of Vienna, Sensengasse 2a, 1090 Vienna, Austria; 2Clinic of General, Special Care, and Geriatric Dentistry, Centre of Dental Medicine, University of Zurich, Plattenstrasse 11, 8032 Zurich, Switzerland

**Keywords:** molar incisor hypomineralization, etiology, structural anomaly, pediatric dentistry

## Abstract

Background: Molar incisor hypomineralization (MIH) is a developmental enamel defect that primarily affects the first permanent molars and sometimes the incisors. Its increasing prevalence worldwide has raised clinical concerns, yet its exact cause remains unknown. This study aimed to assess potential factors influencing MIH development by analyzing the medical history of children aged 6 to 12 years using a questionnaire. Methods: This study included 100 children aged 6–12 years diagnosed with MIH during dental examination, and 100 age-matched children in the non-MIH (healthy) group from the Department of Pediatric Dentistry, University Clinic of Dentistry, Medical University of Vienna. The parents of the participants completed a two-page questionnaire regarding possible etiological factors of MIH. Results: The data analysis involved 100 children with MIH (mean age 8.5; ±1.3; 52% female) and 100 children in the healthy group (mean age 9.2; ±1.3; 42% female). The optimized binary logistic regression analysis revealed a significant association between MIH development and cesarean-section delivery (OR = 3; CI = [1.5–6.2]) and sixth disease (roseola) (OR = 3.5; CI = [1.5–8.0]). Conclusions: This study suggests that cesarean-section delivery and sixth disease (roseola) might increase the likelihood of MIH development in children.

## 1. Introduction

Molar incisor hypomineralization (MIH) is a prevalent dental condition affecting children worldwide [1]. It is characterized by structural enamel hypomineralization of the permanent first molars and incisors, resulting in distinct opacities ranging from white to yellow-brown. Beyond aesthetic impediments, MIH often leads to hypersensitivity and increased susceptibility to dental caries [2]. The global prevalence of MIH varies significantly [1], with regional studies reporting rates of 11% and 7% in Austria, while Germany experiences a notably higher prevalence of up to 30% [3,4,5]. Despite MIH’s status as a global health concern, the details concerning the etiology of MIH remain elusive. Numerous studies delving into potential etiological factors have so far underscored the multifactorial nature of the condition [6,7], which has been linked to prenatal, perinatal, and postnatal factors.

The significance of understanding the prevalence and characteristics of MIH extends far beyond the realms of dental research. This condition carries profound implications for the oral health and overall quality of life of the affected individuals. Thus, it becomes paramount to grasp the multifaceted nature of MIH, not only for the benefit of the affected children but also to inform the development of effective public health strategies and interventions, with a particular focus on pediatric dental care. The prevalence and characteristics of MIH serve as pivotal factors in unraveling the intricacies of this condition. This understanding is the bedrock upon which effective public health policies, targeted treatments, and educational initiatives can be built. By gaining a comprehensive perspective on the scope and nuances of MIH, practitioners and policymakers can be informed with the essential insights needed to mitigate the prevalence of this condition. The determination of these etiological factors is not merely an academic pursuit but an imperative for shaping the future of oral health. Through policies, treatments, and educational efforts, the prevalence of MIH might be influenced.

This study thus aims to contribute to existing research on the etiology of MIH by corroborating existing evidence on previously assumed etiological connections as well as looking at new, previously undetected associations. This is especially important since so far there have been no studies conducted in Austria on the etiology of MIH. The questionnaire used investigates the anamnesis of affected children and their mothers, checking for significant correlations between salient aspects of the patients’ medical history, prenatal and perinatal conditions, maternal health and habits during pregnancy, early childhood nutrition, and exposure to environmental factors.

## 2. Materials and Methods

### 2.1. Study Population

In the present study, a total of 100 children with MIH (test group) aged 6–12 years and 100 children without MIH or any other structural anomalies were included. The non-MIH group consisted of children that were comparable to the test group in terms of age and gender. To be included in the MIH group, the presence of at least one permanent molar affected by MIH, with or without the involvement of incisors, was necessary. All children included in this study were patients of the Department of Pediatric Dentistry, University Clinic of Dentistry, Medical University of Vienna. Written consent to participate in this study was obtained for each patient and signed by their legal guardian. All procedures were conducted in accordance with the Declaration of Helsinki and the study was approved by the Ethics Committee of the Medical University of Vienna (2358/2020).

### 2.2. Examinations

Prior to the commencement of data collection for this study, an initial comprehensive exploration was undertaken by a panel of three highly skilled and proficient pediatric dentists. These examiners represented a select group of dental professionals with specialized expertise in structural anomalies such as molar incisor hypomineralization and have undergone extensive and specialized training in this field. The examination process was executed with the utmost care and precision, ensuring the highest standards of dental assessment. Each participant’s dental health was thoroughly evaluated. The examination environment was optimized to ensure the comfort and ease of the young participants. All children were carefully and comfortably positioned in a chair, facing a well-lit light source that facilitated a clear view of their dental structures. This optimal setup not only allowed for a comprehensive examination but also ensured that the participants’ experience was as pleasant as possible. The criteria established by the European Academy of Pediatric Dentistry were diligently adhered to throughout the examination [8].

### 2.3. Questionnaire

A comprehensive review of the scientific literature was conducted to identify potential etiological factors hypothesized to be associated with MIH. After a thorough examination of the existing evidence and research, a well-structured questionnaire containing 16 questions was developed, targeting the mothers of all participating children. It was designed using reference material, papers, and guidelines on MIH provided by the European Association of Pediatric Dentistry (EAPD). The questionnaire was developed by experts in the field of pediatric dentistry and structural anomalies. This group tested the content and validity of the questionnaire and assessed the accuracy of the questions. Based on the comments made by the selected experts, some questions were added and modified to enable better understanding. The administration of the questionnaire took place at the Department of Pediatric Dentistry, University Clinic of Dentistry Vienna. The questionnaire contained three sections that were divided into prenatal, perinatal, and postnatal conditions. Information regarding the pregnancy, birth, and delivery methods as well as childhood diseases and medication intake or any observed abnormalities during these three periods was gathered.

### 2.4. Statistical Analysis

Only data of children for whom the questionnaires were filled out completely were included in the analysis. The analysis of the data was conducted using the Statistical Package for the Social Sciences (SPSS) software, specifically version 22.0, developed by IBM and headquartered in Chicago, IL, USA. This software was chosen for its robust capabilities for handling complex statistical analyses and its wide acceptance in the field of research. To ascertain the presence of any meaningful relationships between molar incisor hypomineralization (MIH), which was treated as the dependent variable, and the various potential risk factors, considered as independent variables, a combination of well-established statistical tests was employed. This methodological approach was chosen to ensure a comprehensive examination of the data and to extract valuable insights. Furthermore, this study examined the dental health of the participants by assessing the Decayed, Missing, and Filled Teeth (DMF/T) index, which is a critical measure of dental caries. In the initial phase of analysis, Pearson’s Chi-square and Fisher’s exact test were used. These statistical tools, renowned for their efficiency in evaluating categorical data, were skillfully applied to determine the presence of statistically significant relationships between MIH and the potential risk factors. Their utilization allowed us to establish whether these variables were interrelated and to what extent. For a more in-depth investigation, focusing on the independence of significant predictors for MIH, multivariate analysis came to the forefront. Specifically, multivariate logistic regression was the chosen method. This advanced statistical technique is particularly valuable in the context of this research, as it allowed us to explore how multiple independent variables collectively impact the dependent variable, MIH, while controlling for the effects of each variable. By doing so, it was possible to discern the unique contributions of these factors to the likelihood of developing MIH. Further scrutiny and meticulous examination of the data involved the application of Fisher’s exact test and the χ^2^ test. These analyses were carried out based on the construction of contingency tables, which form the bedrock of categorical data assessment. Fisher’s exact test, in particular, is invaluable when dealing with small sample sizes or when expected cell frequencies are notably low. The χ^2^ test, a widely recognized tool, was applied to determine the existence of statistically noteworthy distinctions between the variables. The results of these tests added depth and granularity to the analysis. Upon identifying those variables that exhibited a statistically significant impact on the dependent variable (MIH), a more in-depth approach through exploratory binary logistic regression was taken. This method served as the primary baseline model, enabling an intricate understanding of how these significant factors individually contribute to the likelihood of developing MIH, effectively isolating their unique effects. In addition, refinement was accomplished by means of an optimized, factor-reduced binary logistic regression, integrating prominent *p*-values and pre-established pivotal factors identified from the initial baseline model. In all analyses, a predetermined probability of error was set at an alpha level of 5%. Thus, *p*-values below the threshold of 0.05 were deemed indicative of statistical significance. These comprehensive analyses served to uncover the complex relationships between MIH and potential risk factors, both collectively and individually.

## 3. Results

This study included a diverse cohort of 200 children, ranging in age from 6 to 12 years, in order to comprehensively investigate the factors associated with molar incisor hypomineralization. These children represented a convenience sample which included every patient that was visiting the department that was either diagnosed with MIH or with no structural abnormalities at all. Within this cohort, there was a balanced distribution of gender, with 53% of the participants identifying as male, totaling 106 individuals, and 47% as female, comprising 94 participants (as presented in Table 1). Upon a closer look at the whole sample, it was observed that approximately half of the participants in the MIH group, specifically 52%, were female, while the non-MIH group exhibited a slightly lower proportion of females, accounting for 42% of the participants. The age distribution within the study participants revealed an average age of 8.5 years, with a standard deviation of ±1.3, suggesting a relatively homogeneous age distribution. In contrast, the non-MIH group exhibited a marginally higher average age of 9.2 years, also with a standard deviation of ±1.3. This difference in age between the MIH and non-MIH groups was considered during the subsequent data analysis to ensure that age-related factors were appropriately accounted for. In the MIH group, the mean DMF/T index was calculated at 1.02, with a standard deviation of ±1.4. However, the non-MIH group displayed a slightly lower mean DMF/T index of 0.77, accompanied by a standard deviation of ±1.5. These values were indicative of the dental health status of the participants and served as important parameters for the subsequent analyses. The examination of the extent and severity of MIH revealed intriguing insights. The data analysis, visually depicted in Figure 1, demonstrated that among the children in the MIH group, 51% (n = 51) had affected molars, without any impairment of their incisors. In contrast, 49% (n = 49) of the children in this group displayed hypomineralization in both molars and incisors. Importantly, statistical analysis revealed that this observed distribution did not yield a significant difference (*p* = 0.643), underscoring the complex and multifaceted nature of MIH presentation among the study participants.

In an endeavor to comprehensively investigate the potential etiological factors associated with the presence of molar incisor hypomineralization, this study systematically conducted hypothesis testing, as showcased in Table 2. Firstly, Fisher’s exact test (f) was used to scrutinize the possible perinatal factor of cesarean delivery. The results revealed a statistically significant association (*p* = 0.002), hinting at a conceivable link between the mode of delivery and the presence of MIH. This observation raises intriguing questions about the perinatal elements contributing to this dental condition. Furthermore, the Chi-square test (c) was judiciously employed to assess the influence of breastfeeding duration, a postnatal factor, on the presence of MIH. The outcomes of this analysis were remarkable, displaying an exceptionally significant association (*p* < 0.001). Specifically, breastfeeding for no longer than 12 months could influence the development of MIH. However, further testing with logistic regression did not confirm a significant correlation with the duration of breastfeeding. Additionally, the role of a three-day fever as a postnatal factor was scrutinized using the Mann–Whitney U test (u). The results of this analysis displayed a statistically significant association (*p* = 0.009), underlining the potential impact of postnatal febrile episodes on the development of MIH. The significant associations identified between cesarean delivery, breastfeeding duration, and postnatal three-day fever with the presence of MIH underscore the complex and multifaceted nature of this structural anomaly.

According to the full logistic regression model, cesarean delivery (*p* = 0.008; OR = 3.6; CI = [1.4–9.1]), three-day fever (*p* = 0.006; OR = 4.7; CI = [1.6–14.4]), and otitis media (*p* = 0.034; OR = 3.1; CI = [1.1–8.8]) showed statistically significant associations with MIH development (Table 3). Regarding gender, no statistically significant differences were observed between male and female participants. The goodness of fit of the full logistic regression models was confirmed by Nagelkerke [9] (*p* = 0.538), with 79.80% of the data being correctly classified.

Furthermore, an optimized regression model was developed, considering salient *p*-values and prior important factors from the primary base model (Table 4). This optimized binary logistic regression analysis indicated that cesarean delivery (*p* = 0.002; OR = 3; CI = [1.5–6.2]) and three-day fever (*p* = 0.003; OR = 3.5; CI = [1.5–8.0]) were significantly positively correlated with MIH. All other etiological factors examined did not show significant associations with MIH.

## 4. Discussion

The aim of the present study was to explore possible associations between the etiology of MIH and possible prenatal, perinatal, and postnatal factors. This work is the first in Austria that investigates the aforementioned associations and thus reduces an important gap in this regard. The findings of our study were largely in agreement with previous research in identifying cesarean delivery, limited breastfeeding duration, and three-day fever as factors that increase the likelihood of children developing MIH, while failing to corroborate Vitamin D deficiency and antibiotic intake as etiological factors [10,11,12,13]. Gender analysis did not uncover any significant difference in the occurrence of MIH, which is consistent with the research findings of previous studies from Spain, Finland, and Turkey [13,14,15].

Looking more closely at specific tooth susceptibility, our results (Figure 2) showed that tooth 36 (19.1%) was the most commonly affected of all teeth. According to a cross-sectional study published in 2016, tooth 36 (5.7%) was also the first permanent molar that was most affected by MIH [16].

The *p*-value (<0.001) of the DMF/T index showed that children with an existing MIH have a higher caries risk than study participants from the non-MIH group. Another study that included 840 children aged 6 to 8 years from Valencia reported that DMF/T levels were significantly higher in MIH participants than in unaffected children [15].

Regarding the various possible etiological factors, the most discussed potential prenatal factors, such as compromised maternal health in the last trimester of pregnancy or maternal medication use, were also investigated in the present study. Existing meta-analyses suggest that the likelihood of MIH development is associated with maternal health status. Children whose mothers suffered from health problems during pregnancy were 40% more likely to develop MIH [17]. In this regard, gestational diabetes, drug use, and hypercalcemia have been mentioned as possible prenatal contributing factors in the literature, as well as urinary tract infections in the mother’s last trimester [18,19]. Our study could not show any association between the most commonly studied prenatal factors and MIH, which, although in conflict with the aforementioned studies, does agree with the results of other studies [12,20]. One explanation for this might be the utilization of questionnaires that required participants to recall past events that happened at least 6 years prior to participation in the study. Here, lapses of memory might mean that the actual rate of prenatal disease is greater than the one we observed. 

It is argued that perinatal oxygen deprivation during birth could damage ameloblasts and thus lead to structural anomalies like MIH [21]. In our study, no significant correlation was found between reported oxygen deprivation and MIH development. However, cesarean delivery showed statistical significance in the MIH group. The correlation between cesarean delivery and MIH development has been confirmed in several previously published studies [7,17,22]. Postnatal influences are thought to be of greater importance, as the time frame between the development and eruption of most teeth is mainly postnatal. The analysis in the present study showed that breastfeeding might have an influence on MIH development if the period of breastfeeding did not exceed 12 months [7]. However, no significant correlation between breastfeeding duration and MIH could be confirmed by logistic regression, which is consistent with studies by Laisi et al. and Garot et al. [7,23].

In addition to known general health risks, a link between toxic effects of plasticizers (bisphenol A) and mineralization disorders has been demonstrated in animal studies [24]. Bisphenol A is found in several products that see daily use, such as plastic bottles and pacifiers [25]. In our study, prolonged drinking from plastic bottles and pacifier use did not show any correlation in children with MIH.

Childhood diseases have been frequently mentioned as possible causes of MIH in the literature [7]. Out of the infectious diseases that were investigated in the present study, only three-day fever was confirmed as a potential postnatal factor for MIH development. As of now, the correlation between this infectious disease and MIH has not been investigated in any previous study. However, the results might be biased as the diagnostic difference between three-day fever that is caused by the herpes virus and a common fever infection can easily be confused by many parents [26]. Furthermore, high fever is an accompanying symptom in many respiratory infections [27]. Therefore, the disease itself could influence an occurrence of MIH rather than the accompanying fever [12].

Ear infections are also suspected as another possible cause of MIH development. In our study, middle ear infections had some effect on the occurrence of MIH, which was verified by binary logistic regression analysis. The results of a study by Woullet et al. support an increased risk of MIH occurrence in children who had more than one episode of acute otitis media in the first year of life [14].

One of the most discussed topics in the context of MIH etiology is the possible influence of antibiotic use in children in the first years of life. A cross-sectional study from 2014, which included 267 children, showed a significant correlation between antibiotic use in the first four years of life and MIH [28]. Previously, in 2009, Laisi et al. reported that amoxicillin and erythromycin administration during the first few years of childhood may be associated with MIH [23]. In another retrospective review from Lebanon, researchers found that children who took antibiotics were two times more likely to have MIH than those who were not given any antibiotic therapy [29].

Despite the examination of data in the current study, it is noteworthy that the findings did not reveal a statistically significant correlation between the use of antibiotics and MIH. This observation invites a closer examination of the possible implications and underlying factors at play in this context. One plausible interpretation of these results is that the role of antibiotic use in the development of MIH may be less significant than previously assumed. This raises questions about the conventional knowledge regarding antibiotics as a primary etiological factor in the context of MIH. It prompts us to consider whether other, perhaps more influential, factors contribute to the onset of this dental condition. Are the antibiotic medications to blame, or the disease itself that necessitates the administration of antibiotics in the first place [18]? This consideration underscores the intricate relationship between cause and effect, challenging the conventional assumptions about the relationship between antibiotic medications and MIH. Another aspect to consider is the potential impact of recall bias on this study’s outcomes. Gathering data related to etiological factors, such as antibiotic use, relies on the memory recall of participants and their parents. This reliance on memory may introduce a degree of uncertainty and potential variability in the data. It is essential to acknowledge that relying on the memory of study participants may lead to underestimating the significance of certain results. Memory recall, particularly in the context of healthcare experiences, can be influenced by various factors, including the passage of time and the subjective perception of events. 

In light of these reflections, the non-significant correlation observed between antibiotic use and MIH in this study opens a path for further exploration and inquiry. It emphasizes the complexity of dental etiology and the multifaceted factors that may contribute to the development of MIH.

One significant limitation pertains to the validity and reliability of the questionnaire used in this study. While the questionnaire was designed using reference material, papers and guidelines on MIH and developed by experts in the field of pediatric dentistry and structural anomalies, it is important to note that it requires further testing and validation through further studies. Despite this limitation, our study contributes to the understanding of MIH and provides valuable preliminary data that can guide future investigations.

## 5. Conclusions

The present study demonstrates that cesarean delivery and three-day fever increased the likelihood of the occurrence of molar incisor hypomineralization of affected children in Austria. Future research may benefit from conducting additional validation and reliability assessments. This opens the scope for further longitudinal studies, which circumvent limitations connected to memory.

## Figures and Tables

**Figure 1 dentistry-12-00044-f001:**
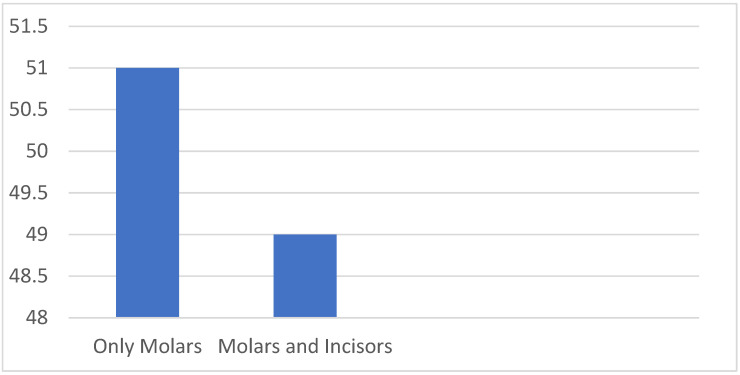
Ratio of MIH-affected teeth in the MIH group in %.

**Figure 2 dentistry-12-00044-f002:**
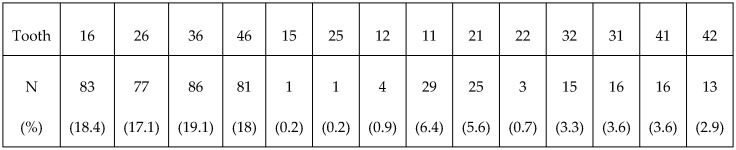
Distribution of all MIH-affected teeth.

**Table 1 dentistry-12-00044-t001:** Summary of sample data (n = 100 patients per group).

	MIH Group	Non-MIH Group
**Gender (%)**		
Male ♂	48	58
Female ♀	52	42
**Age**		
Mean	8.5 years	9.2 years
**DMF/T index**	1.0 (+/−1.4)	0.8 (+/−1.5)
**dmf/t index**	4.5 (+/−3.9)	1.8 (+/−2.6)
**Affected teeth**		
Minimum	1	
Maximum	10	

**Table 2 dentistry-12-00044-t002:** Distribution of etiological factors.

Question	*p*-Value
Did you have any health problems at the end of pregnancy?	0.165
Did you take antibiotics in the last 2 months of pregnancy?	0.369
Did you vomit during the last 2 months of pregnancy?	0.621
Was your baby born premature (before 37 week)?	1.000
Was your child underweight at birth (under 2500 g)?	0.369
Did your child suffer from oxygen deficiency during birth?	0.369
Did you deliver your child by cesarean section?	**0.002 ***
Up to what age did you breastfeed your child?	**<0.001 ***
Did you feed your child with plastic bottles?	0.527
Did your child have a pacifier?	1.000
Did your child suffer of…	
…Rubella?	1.000
…chickenpox?	0.881
…Measles?	0.497
…Three-day fever?	**0.009 ***
…Bronchitis?	0.839
…Asthma?	1.000
…Pneumonia?	0.435
…otitis media?	0.104
…Urinary tract disease?	0.058
Did your child have any febrile illnesses (>38.5 degrees)?	0.247
**Has your child taken antibiotics like…**	
…penicillin?	0.357
…amoxicillin?	0.761
…macrolides?	0.497
…cephalosporins?	0.529
Does your child have a known vitamin D deficiency?	0.065
**Caries index**	
**dmf/t**	**0.001 ***
**DMF/T**	0.092

* considered as significant *p*-Value: <0.05.

**Table 3 dentistry-12-00044-t003:** Binary logistic regression analysis of potential etiological factors.

Question	*p*-Value	Odds-Ratio	Coefficient
Did you deliver your child by cesarean section?	**0.008 ***	3.6 [1.4–9.1]	**1.3**
**Did your child suffer of…**			
…Three-day fever?	**0.006 ***	4.7 [1.6–14.4]	**1.6**
…middle ear infection?	**0.034 ***	3.1 [1.1–8.8]	**1.1**
**Gender**			
Female	0.139	1.9 [0.8–4.4]	0.6
Male	0.265	2.3 [1.2–5.3]	0.8
**Pseudo R2**	**Value**		
Nagelkerke	**0.539**		
**Proportion of correct classification**			
(%)	**79.80**		

* considered as significant *p*-Value: <0.05.

**Table 4 dentistry-12-00044-t004:** Optimized binary logistic regression analysis of potential etiological factors.

Question	*p*-Value	Odds Ratio	Coefficient
Did you deliver your child by cesarean section?	0.002 *	3.0 [1.5–6.2]	1.1
Did your child suffer of…			
…Three-day fever?	0.003 *	3.5 [1.5–8.0]	1.2

* considered as significant *p*-Value: <0.05.

## Data Availability

The raw data supporting the conclusions of this article will be made available by the authors on request.

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
