# Peer review of "Possible Etiological Factors for the Development of Molar Incisor Hypomineralization (MIH) in Austrian Children"

_dentistry, 2024, doi:10.3390/dj12030044_

Round 1

Reviewer 1 Report

Comments and Suggestions for Authors

Author Response

Possible etiologic factors for the development of molar incisor hypomineralization (MIH) in Austrian children           

We thank the editors and reviewers for their thoughtful and helpful comments and suggestions. Please find enclosed our detailed point-by-point response.

We have revised the manuscript accordingly and have indicated all changes in red. We feel our manuscript has substantially improved as a direct result of these comments and hope the revised version is suitable for publication.

Reviewer #1 Comment 1:  No information about the sample size calculations, is it statistically significant?

Response:  We thank Reviewer #1 for this comment.

The study was conducted at the Medical University of Vienna in the Department of Pediatric Dentistry. In Austria we are the only facility that is specialized in the disease Molar-incisor-hypomineralization, thus a main part of our patients is affected by this condition. Rather than selecting a sample from this population, we have the opportunity to include the entire accessible population that meets these criteria. In such cases of complete population inclusion, sample size calculations become unnecessary, as you intend to study everyone within the defined population. Furthermore, we are merely reporting the prevalence of MIH in the population we have. However, Reviewer #1 was right to point this out and we thank them for giving us the chance to respond to them.

Reviewer #1 Comment 2: The author claimed that the questionnaire of this study was valid. However, the validity and the reliability tests were not mentioned.

Response: We thank you for this valuable comment. We indeed failed to report, with sufficient clarity, on how the validity of the questionnaire that was used in our study was assessed. It was designed using reference material, papers and guidelines on MIH provided by the European Association of Pediatric Dentistry (EAPD). The questionnaire was developed by experts in the field of pediatric dentistry and structural anomalies. This group tested the content and validity of the questionnaire and assessed the accuracy of the questions. Based on the comments made by the selected experts, some questions were added and modified to enable better understanding. While we did not conduct additional validation tests in our specific sample, we relied on the validation conducted by the experts in the original development of the questionnaire.

To address these concerns, we plan to take the following steps in our revised manuscript: 

-Clearly acknowledge the limitations regarding the lack of detailed validation and reliability calculations for the questionnaire.

-Suggest that future research may benefit from conducting additional validation and reliability assessments in the specific population or context of our study. This issue has now been clarified in the Material and method section.

We believe that these adjustments will provide a more transparent representation of our study and its reliance on an externally validated questionnaire. We appreciate your valuable feedback and are committed to enhancing the quality and clarity of our research. 

Reviewer #1 Comment 3: Why did you use test and control groups? This is not an experimental study nor an observational prospective one.

Response: Thank you for your valuable feedback. We appreciate your attention to detail and would like to clarify the terminology used in our study.  Our study is not of an experimental or observational prospective nature, as you correctly pointed out. Instead, it is a questionnaire-based analysis of the prevalence of Molar-Incisor Hypomineralization (MIH) among a group of individuals. We understand that the terms "test" and "control" groups, commonly associated with experimental research, may not be appropriate in this context. To better describe our study design, we propose the following terminology: 

-MIH-Affected Group: This term refers to the group of individuals who have been diagnosed with or are suspected to have MIH based on our questionnaire analysis.

-Non-MIH Group: This represents the group of participants who do not have MIH.

We apologize for any confusion caused by the previous terminology, and we will make the necessary revisions in our paper to accurately reflect our study's nature and methodology.  We appreciate your feedback, and we hope that these clarifications help to better convey the essence of our study.

Reviewer #1 Comment 4: dmf/t index 1.8 (+/-2.6) 4.5 (+/-3.9) the control group has more caries than in the test group, this may skew the results.

Response: We thank you for your comment. The observed difference in caries prevalence between the two groups is indeed noteworthy and warrants discussion. However, it is crucial to emphasize that our study primarily focused on the presence or absence of Molar-Incisor Hypomineralization rather than caries as the primary outcome of interest. The patients were chosen based on their MIH status, not their caries status. We would like to stress that, based on our study design and analyses, caries does not appear to influence the presence of MIH or any other results in our study.

­

Reviewer 2 Report

Comments and Suggestions for Authors

I would like to thank the authors for sharing this interesting study that aimed to correlate between the possible etiologic factors for the development of molar incisor hypomineralization (MIH) in Austrian children. Some comments and suggestions for clarification are listed below.

1-    Introduction section:

a.     P2, Ls 44-50: This part looks like methodology carried out by the investigators. Please remove it from the introduction section.

b.    The authors did not state their prespecified hypothesis at the end of this section.

2-     Materials and Methods:

a.     P2, L 53: the authors mentioned that their patients were aged 6-10 years old. This is different than the abstract and the results sections. Please make sure to write the correct range.

b.    100 children with MIH (test group) aged 6-10 years and 100 children without MIH (control group) or any other structural anomalies were included. How was this sample size reached? Please state how your sample size was calculated?

c.     P2, L 64: The authors mentioned that (exploration was conducted by three proficient pediatric dentists before the start of data acquisition for the study). Were the investigators calibrated? Please write the inter and intrarater reliability test values.

d.    The design of the questionnaire is of utmost importance to ensure accurate data is collected so that the results are interpretable and generalizable. In P2, L 73: the authors stated that (a well-structured and validated questionnaire containing 16 questions was developed); however, they did mention the steps of development including translation if any, pilot testing, validity testing …etc. Kindly mention.

e.     It is advisable to revise the STROBE checklist and report your manuscript accordingly.

3-    Discussion section:

a.     Was the recall bias the only limitation for this study?

b.    Please mention whether you accepted or rejected the postulated hypothesis in this section.

4-    Conclusion section:

a.     P 7, L 238,239: (Because the current work is a retrospective study, there are certain limitations in relying on the statements and recollection of the study participants for data collection). Please remove this part. This is not a conclusion; it is a limitation and was mentioned before.

Author Response

Austrian children           

We thank the editors and reviewers for their thoughtful and helpful comments and suggestions. Please find enclosed our detailed point-by-point response.

We have revised the manuscript accordingly and have indicated all changes in red. We feel our manuscript has substantially improved as a direct result of these comments and hope the revised version is suitable for publication.

Reviewer #2 Comment 1:  100 children with MIH (test group) aged 6-10 years and 100 children without MIH (control group) or any other structural anomalies were included. How was this sample size reached? Please state how your sample size was calculated?

Response: We thank you for this valuable comment. The study was conducted at the Medical University of Vienna in the Department of Pediatric Dentistry. In Austria we are the only facility that is specialized in the disease Molar-incisor-hypomineralization, thus a main part of our patients is affected by this condition. Rather than selecting a sample from this population, we have the opportunity to include the entire accessible population that meets these criteria. In such cases of complete population inclusion, sample size calculations become unnecessary, as you intend to study everyone within the defined population. Furthermore, we are merely reporting the prevalence of MIH in the population we have. However, Reviewer #3 was right to point this out and we thank them for giving us the chance to respond to them.

Reviewer #2 Comment 2: The design of the questionnaire is of utmost importance to ensure accurate data is collected so that the results are interpretable and generalizable. In P2, L 73: the authors stated that (a well-structured and validated questionnaire containing 16 questions was developed); however, they did mention the steps of development including translation if any, pilot testing, validity testing …etc. Kindly mention.

Response: We thank you for this valuable comment. We indeed failed to report, with sufficient clarity, on how the validity of the questionnaire that was used in our study was assessed. It was designed using reference material, papers and guidelines on MIH provided by the European Association of Pediatric Dentistry (EAPD). The questionnaire was developed by experts in the field of pediatric dentistry and structural anomalies. This group tested the content and validity of the questionnaire and assessed the accuracy of the questions. Based on the comments made by the selected experts, some questions were added and modified to enable better understanding. While we did not conduct additional validation tests in our specific sample, we relied on the validation conducted by the experts in the original development of the questionnaire.

To address these concerns, we took the following steps in our revised manuscript: 

-Clearly acknowledge the limitations regarding the lack of detailed validation and reliability calculations for the questionnaire.

-Suggest that future research may benefit from conducting additional validation and reliability assessments in the specific population or context of our study.

This issue has now been clarified in the Material and method section as well as in the conclusion.

We believe that these adjustments will provide a more transparent representation of our study and its reliance on an externally validated questionnaire. We appreciate your valuable feedback and are committed to enhancing the quality and clarity of our research. 

Reviewer #2 Comment 3: Was the recall bias the only limitation for this study?  b.Please mention whether you accepted or rejected the postulated hypothesis in this section.

Response: Thank you for your thoughtful questions regarding the limitations of our study and the status of the postulated hypothesis. We appreciate your engagement with our research. While recall bias is indeed an important limitation that we have duly acknowledged in our manuscript, we wish to clarify that it is not the sole limitation of our study. Therefore, the conclusion section has been adapted accordingly.

Reviewer #2 Comment 4: a. P 7, L 238,239: (Because the current work is a retrospective study, there are certain limitations in relying on the statements and recollection of the study participants for data collection). Please remove this part. This is not a conclusion; it is a limitation and was mentioned before.

Response: We thank Reviewer #2 for this comment. The mentioned part has been removed from the conclusion section.

Reviewer 3 Report

Comments and Suggestions for Authors

Possible etiologic factors for the development of molar incisor hypomineralization (MIH) in Austrian children

The manuscript is well structured and well written. It can be recommended for publication based on some observations: 1. The introduction begins with the definition of MIH. 2. Justify the reason for the study and not just limit yourself to mentioning that there are no studies in Austria. That is, emphasize the importance of studies of this nature and their impact on health. 3. At the end of the introduction, include the hypothesis on which the authors carried out the study. 4. Include sample size details. Parameters used for calculation. 5. Include observer calibration data (Kappa tests). 6. In the questionnaire section, include the main references that were considered. 7. Figure 1 is unnecessary. 8. The discussion should include the limitations of the study and emphasize the main contributions and future work. 9. The conclusion must be limited to the objective of the study.

Comments on the Quality of English Language

Minor editing of English language required

Author Response

Possible etiologic factors for the development of molar incisor hypomineralization (MIH) in Austrian children           

We thank the editors and reviewers for their thoughtful and helpful comments and suggestions. Please find enclosed our detailed point-by-point response.

We have revised the manuscript accordingly and have indicated all changes in red. We feel our manuscript has substantially improved as a direct result of these comments and hope the revised version is suitable for publication.

Reviewer #3 Comment 1: Justify the reason for the study and not just limit yourself to mentioning that there are no studies in Austria.

Response: We thank the reviewer for this comment and your valuable suggestion to further justify the reason for our study. We appreciate your insight and provided a more comprehensive explanation of the study's rationale and significance in the introduction section.

Reviewer #3 Comment 2: At the end of the introduction, include the hypothesis on which the authors carried out the study.

Response: We thank you for your comment. We appreciate your input, and we agree that this addition will enhance the clarity and structure of our manuscript.  In our revised version, we made sure to succinctly present the hypothesis at the conclusion of the introduction, thereby providing a clear and explicit statement of the research question our study aims to address.

Reviewer #3 Comment 3: Include sample size details. Parameters used for calculation. Include observer calibration data (Kappa tests)

Response: We thank the reviewer for this comment. The study was conducted at the Medical University of Vienna in the Department of Pediatric Dentistry. In Austria we are the only facility that is specialized in the disease Molar-incisor-hypomineralization, thus a main part of our patients is affected by this condition. Rather than selecting a sample from this population, we have the opportunity to include the entire accessible population that meets these criteria. In such cases of complete population inclusion, sample size calculations become unnecessary, as you intend to study everyone within the defined population. Furthermore, we are merely reporting the prevalence of MIH in the population we have. However, Reviewer #3 was right to point this out and we thank them for giving us the chance to respond to them.

Reviewer #3 Comment 4: The discussion should include the limitations of the study and emphasize the main contributions and future work.

Response: We thank Reviewer #3 for this comment. We outlined the limitations of our study. The discussion section has been corrected accordingly. We believe that by addressing these aspects in our discussion section, we will provide a more comprehensive and insightful interpretation of our study's outcomes. Your feedback has been very valuable, and we are committed to ensuring that our manuscript effectively communicates both its contributions and its limitations.

Reviewer #3 Comment 5:  The conclusion must be limited to the objective of the study.

Response: We thank you for this comment. We appreciate your feedback regarding the conclusion of our study. Your point about aligning the conclusion with the study's objectives is well taken. We agree that it's crucial to maintain clarity and focus in our conclusion.  In response to your suggestion, the conclusion section has been adapted accordingly.

Reviewer #3 Comment 6:  P2, L 53: the authors mentioned that their patients were aged 6-10 years old. This is different than the abstract and the results sections. Please make sure to write the correct range.

Response: We thank Reviewer #3 for this comment. Thank you for bringing the discrepancy in the age range mentioned in our paper to our attention. We appreciate your diligence in reviewing our manuscript. You are correct and we apologize for the inconsistency in reporting the age range of our patients. To clarify and ensure consistency across all sections of the paper, the correct age range for our study participants is 6-12 years. The mentioned line has been corrected in the paper.

Round 2

Reviewer 1 Report

Comments and Suggestions for Authors

Thank you for your responses to my comments. However, the edited manuscript does not show the required changes.

The (test and control groups) are still there!

No limitations of the study-in specific, the validity and the reliability tests of the questionnaire-were mentioned in the discussion section.

Author Response

Possible etiologic factors for the development of molar incisor hypomineralization (MIH) in Austrian children           

We thank the editors and reviewers for their thoughtful and helpful comments and suggestions. Please find enclosed our detailed point-by-point response.

We have revised the manuscript accordingly and have indicated all changes in red. We feel our manuscript has substantially improved as a direct result of these comments and hope the revised version is suitable for publication.

Reviewer #1 Comment 1:  The (test and control groups) are still there!

Response:  We thank Reviewer #1 for this comment.

Our study is an observational study, as we observe the effects of various risk factors on the development/likelihood of MIH. Specifically, it is a case control study where we have identified participants with MIH (cases) and have compared this cohort with participants without this disease (control). These groups are necessary to determine whether the correlations that are found among the cases group are actually statistically significant with regard to the control group. To prevent the false impression that our cases group has been tested, we have decided to label them as MIH and non-MIH groups.

Reviewer #1 Comment 2: No limitations of the study-in specific, the validity and the reliability tests of the questionnaire-were mentioned in the discussion section.

Response:  We appreciate the reviewer's valuable input and recognize the importance of addressing the limitations of our study. Upon reflection, we recognize that a more comprehensive discussion of the limitations would enhance the quality of our paper. In future works, we will make it a priority to provide a more thorough examination of the tools used, including their validation and reliability. We have added the mentioned limitations into the discussion section.

Reviewer 3 Report

Comments and Suggestions for Authors

In my opinion, the observations and suggestions made by the reviewers previously have been addressed and responded to appropriately. The manuscript is now recommended for publication.

Comments on the Quality of English Language

Minor editing of English language required

Author Response

Thank you for your comments.